# Effect of Environmental and Farm-Associated Factors on Live Performance Parameters of Broilers Raised under Commercial Tropical Conditions

**DOI:** 10.3390/ani13213312

**Published:** 2023-10-25

**Authors:** Gustavo A. Quintana-Ospina, Maria C. Alfaro-Wisaquillo, Edgar O. Oviedo-Rondon, Juan R. Ruiz-Ramirez, Luis C. Bernal-Arango, Gustavo D. Martinez-Bernal

**Affiliations:** 1Prestage Department of Poultry Science, North Carolina State University, Raleigh, NC 27695-7608, USA; gustavoquintana22@gmail.com (G.A.Q.-O.); camilaalfaro2164@gmail.com (M.C.A.-W.); 2Grupo BIOS Inc., Envigado 055420, Antioquia, Colombia; juan.ruiz@grupobios.co (J.R.R.-R.); luis.bernal@grupobios.co (L.C.B.-A.); gustavo.martinez@opav.co (G.D.M.-B.)

**Keywords:** broilers, commercial conditions, artificial neural networks, random forest, machine learning

## Abstract

**Simple Summary:**

In commercial poultry operations, chickens are subjected to different housing, environmental, and management conditions, which can drive the performance results at slaughter age. Although modern poultry housing might incorporate environmental electronic sensors, the data collected are not often analyzed to determine the impact of the environment on broiler live performance results. On the other hand, the variability observed in performance can depend on complex interactions between housing characteristics and management that are difficult to elucidate to understand the potential effects on chicken growth. Gathering and analyzing commercial records could be challenging since the datasets produced are large, without structure, and variable. Consequently, the objective of this study was to evaluate the effects of environmental, management, and housing factors on broiler live performance using statistical and machine learning techniques. Several datasets of different sizes were evaluated. Results indicated the vital importance of temperature control during the first three weeks of live performance and livability. Additionally, the results of random forest and artificial neural network analyses indicated the greater relevance of sex, transportation distance of day-old chicks from the hatchery to the farm, and the farm altitude on the BW and feed conversion ratio of broiler flock live performance parameters.

**Abstract:**

Although temperature, relative humidity, and farm-associated factors are known to affect broiler live performance, data about the impact of these variables under commercial operations are still scarce. This study aimed to evaluate the effect of temperature, relative humidity, a thermal humidity index, management, and farm-associated factors on BW, BW gain, feed conversion ratio (FCR), and mortality of broilers raised to 35 d under commercial tropical conditions. The data analyzed included performance records of Ross 308 AP broiler flocks placed between 2018 and 2020. Environmental monitoring information was obtained from electronic sensors that captured data hourly from 80 flocks in 29 farms. Farm-associated factors were gathered using a survey of 86 farms. Three data analyses were conducted in parallel. Correlation analyses, one-way ANOVA, and machine learning techniques were employed. Results indicated that BW and BW gain were reduced, and FCR worsened (*p* < 0.001) up to 21 d when chickens were mainly exposed to temperatures 2.5 °C lower than the recommended optimums for each age period. At the same time, mortality at 28 and 35 d increased. In conclusion, all farm-associated factors affected chicken live performance. Variable importance analysis indicated that performance results at 14 and 21 d were significant to predict BW at 35. At the same time, sex, distance between the hatchery and farm, and farm altitude accounted for the most significant contributions from the farm-associated factors.

## 1. Introduction

During the last five decades, broiler chickens have been subjected to continuous genetic selection [1] to improve BW and feed conversion ratio (FCR). However, these changes have resulted in energy metabolism and expenditure alterations [2,3,4,5,6], making them very sensitive to environmental conditions [7,8]. Extensive research [9,10,11,12,13,14] has demonstrated the optimum T (thermoneutral zone) for ideal performance at all ages and the physiological consequences of keeping birds under higher (heat stress) or lower (cold stress) than the recommended T.

Under heat stress, Goo et al. [15] indicated that chickens subject to 27.8 °C from 21 to 35 d of age resulted in a final BW of 171 g less than their counterparts raised at 20 °C (1676 vs. 1847 g). Similarly, Awad et al. [16] reported a significant increase in the FCR of 0.06 g:g when chickens were exposed to 34 °C, 6 h daily from 22 to 35 d. A recent meta-analysis [17], including seven trials for BW gain and eleven experiments for FCR, demonstrated that heat-stressed chickens gained, on average, 151.4 g less. At the same time, FCR worsened by 0.17 g/g. On the other hand, several authors [5,6,18] have shown the effects of a low T on broiler performance. Ipek and Sahan [18] indicated that broilers subject to cold stress were significantly 111 g lighter on average from 1 to 5 wk of age, while FCR was higher by 0.36 g/g on average from 1 to 3 wk. Zhou et al. [6] demonstrated that chickens exposed to 16 °C between 14 and 17 d of age presented a BW of 18.84 g less and an FCR of 0.23 g/g more at the end of the trial. Similarly, Mohammadalipour et al. [5] reported an FCR 0.09 g/g poorer between 28 and 42 d (2.12 vs. 2.03 g/g) in chickens raised under a cold T from 21 to 42 d. In the same study, higher mortality due to ascites was detected in chickens raised in a cold T compared to normal conditions.

Studies evaluating the effects of relative humidity (RH) on broiler live performance have shown conflicting results across the years depending on the ambient T [19,20,21]. Thus, different T-humidity indices (THIs) have been developed to account for the variation of both environmental parameters [22,23,24,25,26]. Purswell et al. [27] observed that chickens raised between 49 and 63 d of age under a THI (THI = 0.85 × T_db_ + 0.15 × T_wb_) value of 15.3 °C resulted in the best BW compared to chickens subjected to a THI of 26.2 °C (4547 vs. 3873 g). Concomitantly, FCR was substantially increased during this period (6.00 vs. 3.09 g/g) at this level, compared to treatments from 14.8 to 25.8 °C.

In addition to farm environmental conditions, other factors like the transportation of day-old chicks from hatchery to farm, farm location, and equipment, among others, have been related to suboptimal live performance and high mortality in broilers [28,29,30,31]. In a study conducted by Bergoug et al. [28], broilers were lighter at placement (38.4 vs. 40.0 g), 7 (153.3 vs. 158.8 g), 14 (409.5 vs. 423.0 g), and 21 d (872.7 vs. 890.8 g) when day-old chick transportation time from hatchery to farm increased from 0 to 4, and even 10 h. In other reports, Rachmawati et al. [29] showed that when farm altitude was greater than 700 m above sea level (m.a.s.l.), BW at slaughter age was reduced by 184 g (1697 vs. 1881 g). On the other hand, Garces et al. [32] and Toledo et al. [30] have indicated that litter type may affect broilers’ survival rate, BW, BW gain, and FCR.

In commercial operations, only a few reports [31,33] have described the effects of the house, season, equipment, and factors not conventionally evaluated in broiler production. Even though T and RH are usually monitored daily or hourly in broiler farms, and management factors are highly related to flock performance, data about the impact of these variables under commercial operations are still scarce.

The use of commercial data is always challenging [34,35] due to the large volumes of data, the high variability and multidimensionality, repeated measurements over time (time series data), high correlation and multicollinearity, and the lack of structure. The classical objective of statistical models is to identify the exact effect of a set of variables on another. The unstructured, correlated, and multicollinear data from commercial farms include many uncontrolled and unknown effects. If traditional statistics methodologies do not include those confounding variables, the effects measured for the variables included in the models will be biased. Consequently, the prediction ability is reduced.

In contrast, using modern data analytics and machine learning (ML) tools, it is possible to explore environmental sensor data, cluster using farm traits and categories, and identify causality and variable importance that cannot be determined with traditional statistics [34,35,36]. The ML modeling methods tolerate non-linearity, non-normality, and multicollinearity because little to no assumptions are being made. The ML models are designed to make the most accurate predictions, and methodologies include cross-validation for future utilization of models [34,35]. Confounding effects may not have the same impact in ML because the objective is not to identify the exact effect of a variable on another. The ML modeling aims to identify the most likely value of a dependent variable given a set of predictors. ML methodologies like cross-validations [37] tend to identify if the variables included in the model are affected by unidentified confounder variables. The model selection with the best predictability performance should be the least affected by the intrinsic confounding variables in these datasets [37].

This study evaluated the effect of the T, RH, a THI, growth, management, and farm-associated factors on BW, BW gain, FCR, and mortality of chickens raised to 35 d under commercial tropical conditions.

## 2. Materials and Methods

### 2.1. Database and Statistical Software

Ten datasets were obtained from a broiler integration located in Colombia. One dataset contained live performance parameters from 1347 male and 1353 female Ross 308 AP broiler flocks collected weekly from the placement up to slaughter age in 86 farms related to one poultry integrator company. House sizes vary between 10 × 50 m and 16 × 150 m, and some had two or three floors. However, since not all flocks were processed at the same age between 36 and 45 d, the cutoff point on the data was 35 d. Performance records included BW, BW gain, FCR, and mortality of all flocks placed between 2018 and 2020. The weekly performance was obtained from the averages of random samples of the individual BW of 1% of the flocks and feed intake from the records per house. All flocks were subjected to control feeding across three major poultry production regions in the country. Consequently, feed amounts offered were predetermined and recorded. The regions were labeled as Northwest, Midwest, and East. Groups of farms were fed diets of similar composition since feed formulation details are common to the whole company. Farm-associated conditions were collected using a survey for all 86 farms and are detailed in Table 1 and Table 2.

Only 80 flocks had complete hourly data of continuous environmental monitoring with all electronic sensors between January and July 2020 in 29 farms. Electronic environmental monitoring only started in this company in 2019, and clean data were processed in 2020. Some flocks were eliminated from the dataset because at least one of the environmental parameters recorded did not have all the data or presented several outliers due to faulty electronic sensors. These data corresponded to 38 male and 42 female Ross 308 AP broiler flocks. The other nine datasets mentioned contained the broiler house T and RH records data logged automatically every hour over the entire growing cycle. The electronic sensors and datalogger equipment used were provided by Asimetrix Inc. (Asimetrix Inc., Durham, NC, USA). The T and RH were automatically captured using two sensors located within each broiler house and sent via the internet to a central database administered by the provider. Environmental records were shared for analysis as raw data. Then, data obtained from the two sensors were averaged for each hour and identified for each farm and production region to be later used for analysis.

Data were cleaned, organized, and analyzed using R (R Core Team, 2021) in RStudio (RStudio Team, Boston, MA, USA) and JMP Pro 15 (SAS Institute, Cary, NC, USA). Three parallel analyses were carried out with the data collected.

### 2.2. Data Analysis 1: Performance and Environment

The hourly average environmental records captured in 29 farms during the five weeks of each of the 80 flocks were imported to statistical software and merged by farm, date, and time. This corresponds to 67,200 lines of environmental data for all 80 flocks. Then, the placement date and the date at 35 d were added to the environmental dataset to estimate the age of chickens within each flock. These data were highly variable, and other authors have reported that traditional statistical procedures did not find good relationships with performance [36]. Consequently, we calculated the amount of time (hours) chickens were exposed to non-recommended environmental conditions from hatch up to 35 d or within 840 h. The Aviagen management recommendations [38] for house *T* and *RH* were used as optimum values per age. The ideal range of the *T* was ±1.5 °C of the recommended average values. For each flock, exposure time was calculated to be lower or greater than 1.5, 2.5, 5.0, and 7.0 °C from the recommendations with an *RH* below 50 and above 75%. Additionally, a *THI* was computed for each hour based on the equation described by Berman et al. [39] as follows:THI=3.43+1.058×T−0.293×RH+0.0164×T×RH+35.7
where

*T* = Temperature;*RH* = Relative humidity.

Index boundaries for each day of age were calculated based on the recommended *T* and *RH* combination. Exposure time to *THI* levels out of limits was also considered for analysis. Since feed allowance was controlled during the growth of chickens, this variable was not included in the data analysis.

Correlation analyses were conducted between the BW, BW gain, FCR, and mortality parameters of the 80 flocks with environmental monitoring and hours of exposure to non-recommended *T*, *RH*, and *THI* levels. Since these flocks were subjected to control feeding, feed intake was not considered for these analyses. Correlations were fit using either week or cumulative periods of exposure from placement to each week of age up to 35 d. Linear regressions were carried out on correlated variables with live performance parameters as a response and environmental exposure time as a predictor.

### 2.3. Data Analysis 2: Farm Management Factors and Performance

Farm-associated and management factors of 86 farms were joined to the broiler performance records of 2700 flocks (1347 male and 1353 female flocks) using the farm name as a link. Continuous variables from the farm management survey conducted in 86 farms, like altitude, litter reuse cycle number, and downtime between flocks, were clustered by FCR at 35 d to determine groups to compare. The clusterization method used was the k-means procedure [40] in JMP 15 (SAS Institute, Cary, NC, USA). Several clusters were developed based on an iterative fitting process. Briefly, the k-means procedure algorithm randomly took many starting points as cluster seeds or centers and computed the distance between each pair of datapoints. Then, it iteratively moved the centers to minimize the total within-cluster variance [40]. The k-means cluster with the lowest AIC and BIC was chosen for further analysis. Broiler flocks with missing data related to management conditions or those excluded within the clusterization procedure were not considered for analysis.

After clustering all possible variables, data were analyzed in a one-way ANOVA with variable cluster (continuous variables) or factors (categorical variables) as main effects and BW, FCR, or cumulative mortality at 35 d as responses. Mean separation was performed using the LS means method using Tukey’s or student’s t-test at a significance level of alpha = 0.05.

The management survey indicated that the company was divided into three major poultry production regions: Northwest, Midwest, and East, which received chicks from two company-owned hatcheries (Hatchery 1 and Hatchery 2). Farms also had two technification levels, a high level that accounted for farms with greater capacity, more houses, and more retrofitted houses and a low technification level for farms with less capacity and few houses with open-sided structures. Farms were located up to 531 km or 11 h away from the hatchery. These two variables were clustered into three groups, near (73 ± 71 km), intermediate (356 ± 169 km), and far (454 ± 64 km) for distance in km. Three groups accounted for the distance measured in hours, near (2.1 ± 1.6 h), intermediate (7.0 ± 2.9 h), and far (8.2 ± 1.8 h). Also, farms had an average altitude of 1381 ± 449 m.a.s.l., which were grouped into low (1041 ± 349 m.a.s.l.), middle (1446 ± 498 m.a.s.l.), and high (1670 ± 434 m.a.s.l.) altitude. Poultry houses were equipped with rice hulls or wood shavings as litter material, and the frequency at which the farm recycled the litter was clustered into 0.8 ± 0.7, 1.3 ± 1.2, 5.8 ± 2.2, and 12.0 ± 0.01 times. The downtime between flocks resulted in three ranks, 12.7 ± 1.0, 13.5 ± 1.2, and 15.0 ± 1.6 d.

Other characteristics linked to house infrastructure were also detailed. It was observed that some farms had poultry houses with 1 or 2 stories, which allowed them to place chicks on both floors, and the house floor was composed of two materials—soil or concrete. Additionally, different systems of water storage were reported by farm managers. Water tanks that had any protection against sun and rain (covered), tanks outside the house with no environmental protection (uncovered), and water tanks located underground.

### 2.4. Data Analysis 3: Prediction of Performance with ML

Three supervised ML methods were evaluated in this analysis [35,37]. These methodologies were multiple linear regression (MLR), random forest (RF), and artificial neural networks (ANN). An MLR model was fitted using the lm function from the R stats package. Additionally, an RF and ANN were trained and validated using the caret package [37].

Farm-associated factors were joined to the performance records from all flocks to the model chickens’ BW and FCR at 35 d. BW and FCR at 35 d were used as responses, while performance records and farm-associated conditions were used as predictors to fit these models. Only broiler performance records up to 21 d of age were included to obtain a prediction two weeks before slaughter age.

Data were automatically divided into random training and testing datasets using a five-fold cross-validation procedure to fit the models. Models were then trained using four-folds, and the predictability performance was assessed with the hold-out fold, repeating this step for each of the five folds [37,40,41]. This k-fold cross-validation inevitably results in a lower coefficient of determination (R^2^) but more confidence for predictability [42]. Categorical predictors such as farm region or litter type were transformed into dummy or indicator variables that denote the absence or presence of a factor with 0 and 1, respectively, and data were normalized in a range from 0 to 1 [37,43].

The R^2^ and root mean square error (RMSE) were calculated [37] and averaged across five folds for each model for model evaluation [42]. In the MLR, collinearity was evaluated with the variance inflation factor (VIF). Variable selection was carried out based on the *p*-value. Thus, collinear and non-significant predictors were manually removed to obtain the final model.

A variable importance analysis [44,45] was conducted in all models. This analysis indicates how much a specific variable contributes to the predictive response in a given model to obtain accurate predictions. In general, variable importance is calculated with the relative influence of each variable, and the specific calculation method varies among the three methods evaluated in the analyses described here. In the MLR, the variable contribution was estimated as the predictor percentage sum of squares over the sum of squares of the model. For RF, the permutation importance method [46] was employed to determine variable importance and was expressed as the increase in mean square error (MSE). The ANN utilized the connection weights approach described by Olden et al. [47] to calculate the order of variable importance.

## 3. Results

### 3.1. Data Analysis: Performance and Environment

The house environmental data corresponded to 29 farms and 80 flocks analyzed. Correlation coefficients of live performance parameters and exposure time to the T below or above recommended values were determined. The dataset did not contain flocks exposed to temperatures above 9 °C from the recommended values of the standard genetic line.

The flocks with higher *r* and R^2^ are shown in Figure 1 and Figure 2. Overall, live performance results were mainly correlated (*p* < 0.05) to exposure to the non-recommended T up to 21 d, except for cumulative mortality, which included cumulative exposure values up to 28 d. Only 10 flocks out of the 80 evaluated with continuous environmental monitoring had a T above the target during the first 4 wk of life. No correlations between T above target and BW, BW gain, and FCR (*p* > 0.05) were observed throughout the growing cycle.

Negative correlations of BW (*p <* 0.05) were detected when chickens were exposed to a T 5 °C below the recommended values at 7, 14, and 21 d. Linear regression estimates from the same week of age indicated (*p* < 0.05) that chickens lost 1.51, 3.50, and 4.91 g, respectively, each hour they were exposed to these conditions. Also, cooler environmental T affected FCR at the second and third week of age (*p* < 0.05). Chickens exposed to a T 2.5 °C lower than the recommended from 7 to 14 d or 14 to 21 were less efficient with 0.011 and 0.004 g/g each additional hour under these conditions at 14 at 21 d, respectively. No significant correlations (*p* > 0.05) between a low T and mortality were observed during the first three weeks of age. However, a non-recommended T up to four weeks of age affected cumulative mortality at four and five weeks of age. As time under a T below 2.5 °C from 0 to 28 d increased (*p* < 0.05), mortality augmented on average 0.018% at 28 and 35 d per each additional hour of exposure.

No significant correlations (*p* > 0.05) between BW or BW gain with environmental T was observed at 28 d. FCR significant correlations (*p* < 0.05) with time were exposed to a T below the recommended values during the last week. Positive correlations (*p* < 0.05) and regression analyses indicated that each hour exposed to a T below 2.5 °C up to the fourth week of age reduced the FCR by 0.004 g/g at 35 d.

On the other hand, exposure to a T above the recommended values affected mortality (*p <* 0.05) from 28 to 35 d. Regression analyses resulted in the following equations. Mortality = 0.208 + 0.028 × Hours of exposure (*p* < 0.001; R^2^ = 0.88). It indicates that chicken flocks exposed to 7 °C above the target T from 7 to 14 d of age resulted in a mortality 0.028% greater in the last week of growth for each hour of exposition. Consequently, the same variable similarly affected cumulative mortality at 35 d (0.208% per hour of exposure to T 7 °C above target from 7 to 14 d). No other correlations (*p* < 0.05) or relationships were observed among flock performance variables with the T above target.

Results from correlations and regression analyses between live performance parameters and times of exposure to non-recommended values of RH are detailed in Table 3. Significant correlations (*p* < 0.05) between FCR or mortality and time exposed to non-recommended RH values were detected at 7, 21, and 28 d of age. However, the correlation coefficients estimated were between 0.41 and −0.51 (Table 3). The regression analyses exhibited poor goodness-of-fit with R^2^ lower than 0.26 (Table 3). Similarly, correlation coefficients between live performance parameters and THI levels indicated that only FCR at 21 d was positively associated (*r* = 0.69; *p* = 0.025) with exposure at lower levels of THI. The equation fitted at 21 d was FCR = 1.284 + 0.007 × Hours of exposure (*p* = 0.025; R^2^ = 0.49). This linear regression indicated that the FCR worsened by 0.007 g/g for each hour that chickens were exposed to THI levels lower than the recommended from 14 to 21 d.

### 3.2. Data Analysis 2: Farm Management and Infrastructure Factors and Performance

Significant effects of farm-associated factors (*p* < 0.05) were detected in male and female BW, FCR, and cumulative mortality at 35 d (Table 4). In this case, 86 farms and 2700 flocks were analyzed. Male chickens from the Northwest region of the country resulted in the lowest BW (*p* < 0.001) but, at the same time, the best FCR (*p* < 0.001). In contrast, females from the East region were observed to show the worst live performance with the lightest chickens and the highest (*p* < 0.001) FCR. Cumulative mortality at 35 d in farms from the East region was, on average, 0.39% higher in both males and females (*p* < 0.001) than in farms located in the Midwest (2.78 vs. 2.39%). On the technification level, farms with high levels produced heavier male chickens (*p* < 0.001) but were less efficient (*p* < 0.001) than their counterparts in the low category. Contrarily, low-level farms produced female chickens 18 g heavier (*p* < 0.001) and 0.017 g/g more efficiently (*p* < 0.001) than those raised in high-technification farms.

The live performance of chickens raised at different geographic altitudes also varied (*p* < 0.001). Chickens raised at higher elevation (1670 ± 434 m.a.s.l.) resulted in the best performance with males and females that were 106 and 114 g (*p* < 0.001) heavier, 0.120 and 0.098 g/g (*p* < 0.001) more efficient, with 1.11 and 0.63% (*p* < 0.001) less mortality at 35 d compared to chickens placed on middle-altitude farms (1446 ± 498 m.a.s.l.). Flocks raised in lower altitudes (1041 ± 349 m.a.s.l.) resulted in intermediate responses.

Conflicting results were observed in chickens from different hatcheries (Table 5). Male chickens from Hatchery 1 displayed less FCR (*p* < 0.001) and the lowest BW than chickens from Hatchery 2. Different from males, females exhibited the highest BW (*p* < 0.001) and lowest FCR (*p* < 0.001) when chicks were obtained from Hatchery 1. At the same time, females from Hatchery 2 were 42 g lighter and 0.038 g/g less efficient than their counterparts from Hatchery 1 at 35 d. No significant effects of chick source (*p* > 0.05) were identified in cumulative mortality at 35 d for males or females. Similar results were observed on chick transportation measured in the distance between the hatchery and the farm in km or time (h) needed to reach the farm. Male chicks subject to long trips (454 ± 64 km) were 97 g heavier (*p* < 0.001) at 35 d (1909 vs. 1812 g) than chickens subject to intermediate trips (356 ± 169 km). Still, the male and female chickens raised in farms near (73 ± 71 km) to the hatcheries were significantly (*p* < 0.001) 0.098 and 0.025 g/g more efficient than those chickens traveling 356 ± 169 km (1.473 vs. 1.571 g/g) or 454 ± 64 km (1.473 vs. 1.498 g/g), respectively, to reach the farm. Chickens traveling intermediate distances (356 ± 169 km) presented on average 0.86% and 0.74% more (*p* < 0.001) male and female mortality, respectively, (*p* < 0.001) than chickens from the short (73 ± 71 km) and long 454 ± 64 km clusters.

Differences due to litter type (*p* < 0.05) indicated that male chickens raised in rice hulls were heavier (*p* < 0.001) and had worse FCR (*p* < 0.001) compared to those raised in wood shavings. In contrast, females accounted for lower BW (*p* < 0.001) and worse FCR (*p* < 0.001) than their counterparts placed in wood shavings. Additionally, the number of times the litter was recycled affected (*p* < 0.05) BW, FCR, and mortality at 35 d. Flocks subject to non-recycled litters or with the minimal number of times (0.8 ± 0.7) resulted in males and females on average 80 g and 65 g lighter (*p* < 0.001), respectively, than flocks from all other clusters. Flocks in this cluster of low litter recycling were also (*p* < 0.001) the least efficient (1.556 g/g). They had the highest mortality (*p* < 0.001) for both males (0.63% more) and females (0.74% more) compared to flocks raised in litter recycled among all other groups. Finally, flocks with an average downtime of 13.5 d resulted (*p* < 0.001) in the heaviest (1906 g for males and 1778 g for females) and most efficient (1.457 g/g for males and 1.466 g/g for females) flocks at 35 d with the lowest mortality (2.51%) compared to flocks whose downtime was shorter (12.7 ± 1.0) or longer (15.0 ± 1.6).

### 3.3. Data Analysis 3: Prediction of Performance with ML

***BW Prediction***: The results of the five-fold cross-validation process presented in Table 6 indicated that the RF had the better fit (R^2^ = 0.78), followed by the ANN (R^2^ = 0.73) and MLR (R^2^= 0.59). Hyperparameter tuning [37,42,43] indicated that the best fit of RF was achieved with 37 parameters, while the best fit of ANN included three nodes. On fitting the FCR, goodness-of-fit metrics for five-fold cross-validation resulted in an R^2^ of 0.18, 0.48, and 0.45 for MLR, RF, and ANN, respectively. Variable importance analyses [44,45,46,47] demonstrated that the MLR for BW included only eight non-collinear and significant predictors from which sex accounted for 89% of the total response, feed intake at 14 d with 4.41%, and BW gain at 7 d with 1.82% (Table 6). The RF model obtained for BW indicated that BW at 21 d, sex, and distance from hatchery to farm in km was ranked as the first three most important to predict the final response at 35 d (Figure 3a). Other farm factors were also ranked as critical factors for prediction. These factors included farm altitude (21.98%) and the region where the farm was located (14.05%). Among farm-associated factors, the downtime between flocks presented a 14.72% increase in MSE, becoming the most important factor. Similarly, the ANN showed that mortality at 14 d, BW at 21 d, and BW gain at 21 d were vital in the model (Figure 3b). However, this model detected predictors related to farm infrastructure like water tank location, type of house, and farm area within the top 20 factors to predict BW at 35 d.

***FCR prediction***: In MLR, the FCR at 21 d was the most critical variable, with 42.41% of the total response, followed by downtime between flocks with 18.67%, feed intake at 14 d with 17.63%, and percentage of retrofitted houses (infrastructure) with 7.34%. The other predictors in the model contributed less than 4% each. A total of 10 variables were included in this model. In the variable importance analysis from the RF of FCR at 35 d, the FCR at 21 d was the most important variable (63.26%) compared to other predictors, followed by sex (24.07%) and farm elevation (18.59%) (Figure 3c). Like the BW models, BW, BW gain, and FCR at 14 and 21 d were within the 20 most important factors, and distance between farm and hatchery (16.86%) was listed within the ten first ones. The ANN demonstrated that the response strongly depended on the feed intake, FCR, and BW results at 21 d, followed by the percentage of retrofitted and open-sided houses in the farm (Figure 3d). Again, the BW and BW gain at 14 d and the downtime between flocks are essential factors for predicting FCR at 35 d.

## 4. Discussion

### 4.1. Broiler Performance and Environment

In the dataset explored in this study, no effects of the T above the target were observed, except for mortality between 28 and 35 d. These farms are located in the Andean tropical conditions, which are mountains with microclimates where the T at night and early in the morning always reduces below brooding targets. Considering the low housing insulation and open-sided structures, a low T could be a more prevalent problem than heat stress in these mountain farms.

During the first three weeks of age, cumulative exposure to 2.5 °C below the recommended T correlated to poorer BW, BW gain and FCR, and high mortality during the production cycle. In a recent study, Su et al. [48] demonstrated that chickens exposed to 3 °C lower than the control group from 8 d onwards resulted in similar FCR and BW up to three and six weeks, respectively, compared to chickens kept under thermoneutral conditions. In contrast, chickens exposed to 12 °C lower than the control group from 8 d onwards were significantly lighter and less efficient than their counterparts from two up to six weeks of age.

In another report, Candido et al. [13] indicated that a mild cold T during the early stages (27 °C first week and 24 °C in the second week) did not affect BW and FCR from 0 to 21 d, while the most significant BW gain was observed in chickens subjected to these Ts. According to those studies, 3 °C below the ideal T was still within the versatile T range for chicks, preventing a reduction in the live performance. Although in the current study, no significant correlations of T lower than 1.5 nor 2.5 °C with live performance parameters were detected within the first week of age, the sensitivity of chickens to lower T seemed to increase with age since more exposure to mild cold T worsened FCR at 14 d and BW and FCR at 21 d. Other authors that have reported similar results [2,3,5,6] have indicated that an increase in the basal metabolic rate and energy metabolism due to cold stress led to the rise in chicken energy requirements, which explained the diminished BW or increased FCR observed here. Zhou et al. [6] also suggested that these effects were related to a redistribution of nutrients during the growth toward thermoregulatory responses.

On the other hand, chickens mainly exposed to intermittently T below 2.5 °C from recommendations up to four weeks of age were correlated with high mortality and worsened FCR at 28 and 35 d. Similarly, Bruzual et al. [49] demonstrated that chicks raised to 12 d under cooler brooding T (26 ± 4 °C first week and 24 ± 4 °C second week) presented 1.71% more mortality compared to chicks subject to warmer T (32 ± 4 °C first weel and 29 ± 4 °C second week). Another report indicated that the highest mortality at 42 d resulted from chicks exposed to 26.7 °C in the first week, compared to other treatments (4.79 vs. 2.36%) that ranged between 29.4 and 35 °C [50]. The information obtained from the analyzed dataset suggested that a low T in commercial operations might lead to mortality at the end of the production cycle. It is also necessary to consider that previous reports from the literature described in the present study have been conducted under experimental conditions, achieving desired environmental T and RH through environmentally controlled chambers, which significantly differed from environmental conditions under commercial operations that mainly reflect variability in house environmental conditions.

Additionally, environmental conditions in the tropics may remarkably influence the poultry house environment due to the open-sided structure, representing 90.14% of all houses evaluated in this company. Thus, cycling T between day and night or hourly, precipitations, region, geographic altitude, cold wind drafts from mountains, ventilation and heating systems, and other variables could drive the environmental response observed in this poultry company. It is assumed that mainly the highly variable indoor environmental conditions (Figure 4) resulted in more cumulative hours of exposure to the house T and RH below recommendations. These house variable conditions affected broiler flocks during the first two weeks since it is more difficult for chickens to adapt to variable environments [51], and consequently, live performance at slaughter age was negatively affected.

On fitting the RH data ranged from 30.32 to 92%, only a few mild correlations were observed up to 28 d, while linear models among correlated variables had low R^2^ (0.17–0.26). Similarly, Zhou et al. [21] reported that average daily feed intake, FCR, and mortality were not affected when chickens were exposed to RH of 35, 60, or 85%. In contrast, Weaver and Meijerhof [19] detected broilers to be on average 32 g lighter at 42 d when they were either subjected to weekly increments of 8% of RH from 40 to 80% or raised at a constant 75% RH. Yahav [20] indicated that BW at 35 d was reduced when chickens were reared in environments with RH less or greater than 60–65% compared to other treatments (40–45, 50–55, and 70–75%), while no differences among treatments were observed on FCR or BW at 28 d. However, it was described [20] that the response varied at different ambient Ts (28 or 30 °C). Then, RH could rely on an ambient T to affect live performance. At 28 °C, Yahav [20] determined that the heaviest broilers were observed between 60 and 65% of RH compared to other treatments. Chickens within RH treatments of 50–55% and 70–75% but at 30 °C improved their BW (3.74%), while the 40–45% RH treatment obtained the lightest chickens in both scenarios at 28 (1438 g) and 30 °C (1398 g). Zhou et al. [21] also indicated that although average daily feed intake was not affected by RH treatments at 26 °C, this parameter was reduced by 35 and 85% RH at 31 °C.

THI analyses indicated that as hours of exposure from 14 and 21 d under THI levels below the combined Aviagen recommended T and RH increased, the FCR at 21 d worsened by 0.007 g/g. In contrast, Purswell et al. [27] did not detect effects on live performance parameters between 49 and 63 d when chickens were exposed to a THI between 14.8 and 20.7 °C. The BW, BW gain, feed intake, and FCR diminished in that study with higher THI levels. It is assumed that the well-developed feather cover that those animals could have at 49 d helped to resist lower THI levels and resulted in similar responses at slaughter age. Contrarily, chickens at 21 d from the current study may not have a good development of feathers, which causes exposure to slightly lower THI to increase mortality. According to the results presented herein, BW, BW gain, and FCR of chickens were more affected by a low T during the first three weeks of age. At the same time, mortality was associated with an increase of up to 125 h between 0 and 28 d in the exposure to a T 2.5 °C lower than recommended. The RH between 30.32 and 92% seemed to be a parameter that did not significantly affect the live performance of chickens possible to detect in this data analysis. The THI employed here [38] did not show a significant relationship with performance variables.

Finally, the methodology described herein based on hours of exposure to different environmental conditions can improve the understanding of the effects of T and RH since mean environmental values, maximum and minimum T per day, could not reflect the actual fluctuation within the poultry house. In conclusion, a lower T than recommended affected the live performance of broilers during the whole production cycle. On the contrary, only a few correlations were observed with a T above the target and RH, while the THI did not depict the effects observed with a lower T.

### 4.2. Farm Management and Infrastructure Factors and Performance

On farm-associated factors, evaluating 86 farms and 2700 flocks, chickens that were raised at the highest altitude (1670 ± 434 m.a.s.l.) were the heaviest (1836 g) and the most efficient (1.466 g/g) in both males and females compared to those chickens raised at a medium elevation (1446 ± 498 m.a.s.l.). Rachmawati et al. [29] indicated that the heaviest chickens at 35 d were observed when farms were located above 700 m.a.s.l. In comparison, no differences were detected between chickens raised either in lowlands (<400 m) or middle lands (400–700 m). In that study, the authors suggested that a higher BW could result from a greater feed intake associated with an increment in the maintenance requirements due to a cooler ambient T [29]. In contrast, it is assumed that higher altitudes may represent lower environmental Ts, reducing the effects of heat stress observed in low-altitude regions.

In the present study, the rice hulls as litter type also worsened the FCR of both males and females compared to wood shaving. The litter material like wood shavings, sawdust, sand, rice hulls, wheat straw, and grass in which chickens are raised is a factor that has been widely evaluated in poultry [32,52,53,54,55]. Garcês et al. [32] reported that chickens raised to 35 d in rice hulls did not significantly differ in BW, FI, or FCR compared to those raised in wood shavings. Nevertheless, the survival rate of chickens reared in rice hulls was 4.8% lower than their counterparts reared in wood shavings [32]. Still, the number of replicates in that study was limited to only three pens per treatment, which could not be enough to reproduce the results, while the current data analysis utilized, on average, 601 flocks for each litter type.

Similarly, using two replicates per treatment, Ramadan and El-Khloya [56] showed that live performance parameters or carcass traits were not affected by the litter type when using five different types of litter, including wood shavings and rice hulls. Toledo et al. [30] conducted a comprehensive meta-analysis indicating that broilers raised in wood shavings presented higher BW and better FCR than those reared on rice hulls or alternative litter, respectively. It has been suggested that rice hulls account for the more significant proliferation of fungi and, consequently, mycotoxins [57], worsening live performance in broilers.

In addition, the results presented herein indicated that BW in males and females improved when the litter was recycled more than once. Abougabal [58] observed no significant differences in the BW from 1 to 6 wk of age nor in the FCR at the end of the experiment when evaluating four litter treatments (new litter, 50% new litter–50% reused litter, 100% reused litter, and 100% reused treated litter). Still, the author suggested that recycled litter was not hazardous for broiler production performance. In contrast, an experiment conducted by Garcés-Gudino et al. [59] in tropical conditions indicated that BW (1922 vs. 1753 g), BW gain (53.7 vs. 48.8 g/d), FCR (1.588 vs. 1.633 g/g), and mortality (1.61 vs. 3.2%) at 35 d were improved when the same litter was used for two or three production cycles. It has been mentioned that the recycling process may have beneficial impacts on the performance of broilers since the litter bacterial environment could work as a probiotic or a direct-fed microbial supplementation in chickens that contributes to improving the immunity and the response against coccidia challenges, which usually affect the live performance of broilers [59,60,61,62].

The distance from the hatchery to the farm measured in both kilometers and time spent to reach the farm showed that chicks traveling the longest distances were heavier at 35 d. In contrast, Bergoug et al. [28] indicated that chicks traveling 4 and 10 h had less BW from the placement up to 21 d, where the effect of transportation time disappeared, compared to chicks that traveled less than 5 min to the farm (0 h, control). However, it was further investigated with hatchery shipping reports that the results presented herein could vary because chicks intended to travel long distances are usually scheduled to start the journey at night, which could reduce stress generated by heat, dehydration, and feed and water deprival, in comparison to chicks delivered during the day. Those aspects related to stress during transportation could be more detrimental to chick development and health than the duration of the trip per se [63,64,65,66]. Still, males and females were more efficient when post-hatch transportation was the shortest.

When evaluating these factors together with the growth parameters, the variable importance analyses from the supervised ML models demonstrated that the BW and FCR at 35 d are highly dependent on the live performance that chickens may exhibit during the second and third week of age. Other parameters like sex, the distance between the hatchery and the farm, and farm elevation were important to predict BW.

### 4.3. Prediction of Performance with ML

The MLR presented the lowest predictability for BW and FCR at 35 d. In contrast, the RF and ANN had the best model fit to predict both the BW and FCR at 35 d, with RF being the method with the greatest predictability capacity. It suggests that both RF and ANN could be used to predict performance results in broiler operations. Nevertheless, it is important to consider the pros and cons of each ML technique before scaling to commercial settings. Some studies have demonstrated the use of ANN to predict responses in broiler performance [67], behavior [68,69], and health [70,71]. Although those studies exhibited excellent model predictability performance (R^2^ > 0.99), those studies were conducted using small datasets from experimental units, including a few individuals, and under controlled conditions with specific treatments and pre-established data structures. Those characteristics do not represent the normal variability observed in data from commercial broiler operations. A few studies have explored using RF to predict responses in poultry, mainly for broiler breeders under precision feeding [72,73,74] or using sound data [75]. Only a few reports have used broiler commercial data to predict relations among the variables [34,76], indicating that RF was also the best predictive methodology.

The variation of the FCR was explained in less than 50% with these ML techniques under five-fold cross-validation. Based on the present data, it was concluded that the contribution of management, environment, and infrastructure to the FCR at 35 d is less than 50%. Most likely, its variability is associated with other factors like nutrient intake, which was not included in those models. Finally, BW at 21 d, sex, distance from hatchery to farm, feed intake, and farm altitude were the five most important variables to predict the BW at 35 d. In comparison, FCR at 21 d, sex, farm altitude, feed intake at 21, and FCR at 14 d accounted for the most essential factors in predicting FCR at 35.

## 5. Conclusions

In conclusion, temperatures lower than 2.5 °C added up to the variable environmental conditions reduced the BW and BW gain at 7, 14, and 21 d, while the FCR was worsened at 14 (0.011 g/g), 21 (0.004 g/g), and 35 (0.004 g/g) d of age. Not enough flocks were exposed to temperatures above 1.5 °C in these datasets to detect the effect of heat stress early in life. In addition, mortality increased by 0.018% on average at 28 and 35 d when chickens were exposed to temperatures lower than 2.5 °C. All farm-associated and chick source factors affected the performance of broilers, and sex, distance from the hatchery to the farm, and farm altitude were within the five most important variables to determine broiler BW at 35 d. The intermediate chick transportation distances (356 ± 169 km or 7.0 ± 2.9 h) worsened the final BW, FCR, and increased mortality due to environmental conditions during the hot hours of the day. The best performance was observed in high altitude (1670 ± 434 m.a.s.l.). Wood shavings, instead of rice hulls as the litter type, and litter reuse are positive for broiler flock performance.

## Figures and Tables

**Figure 1 animals-13-03312-f001:**
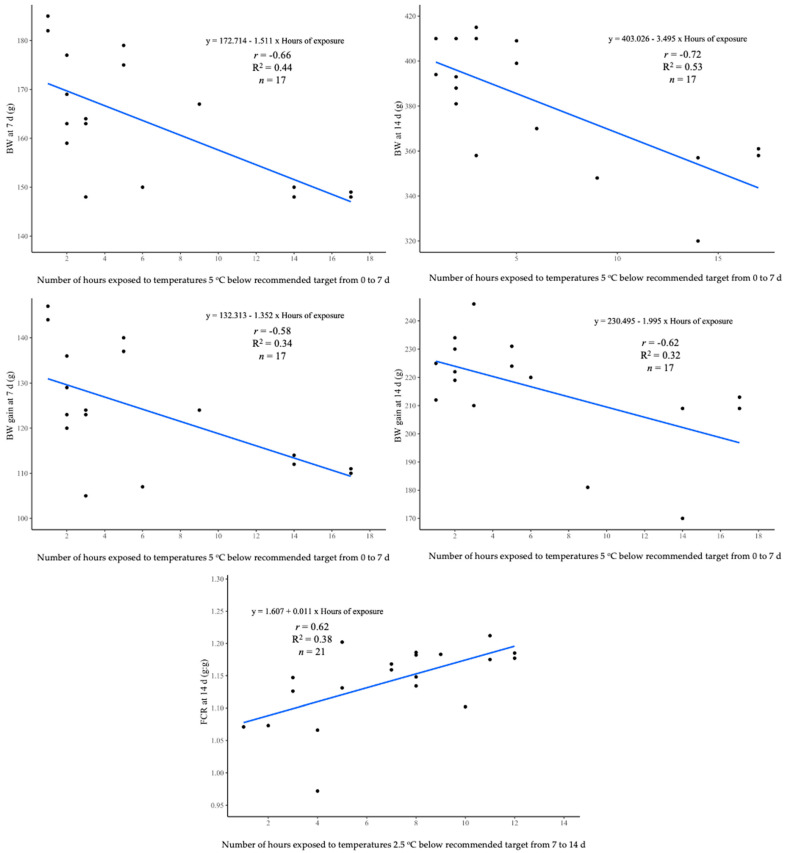
Relationship between the live performance parameters of Ross 308 AP broilers from 7 to 14 d and the hours of exposure to temperatures below the recommended values by Aviagen [38]. *n* = 80 flocks.

**Figure 2 animals-13-03312-f002:**
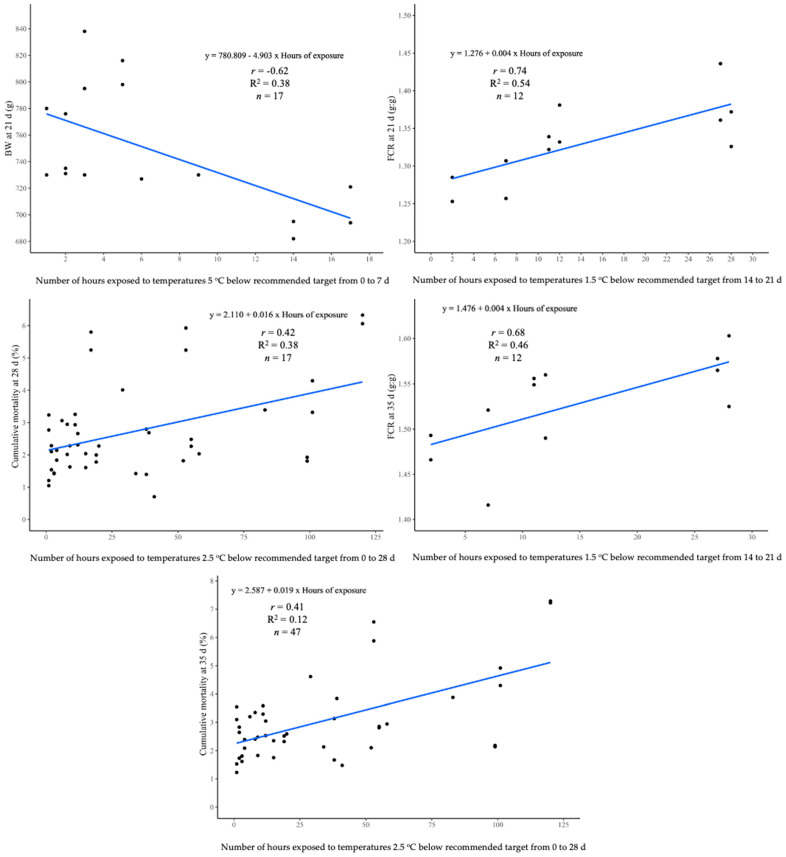
Relationship between live performance parameters of Ross 308 AP broilers from 21 to 35 d and the hours of exposure to temperatures below the recommended values by Aviagen [38]. *n* = 80 flocks.

**Figure 3 animals-13-03312-f003:**
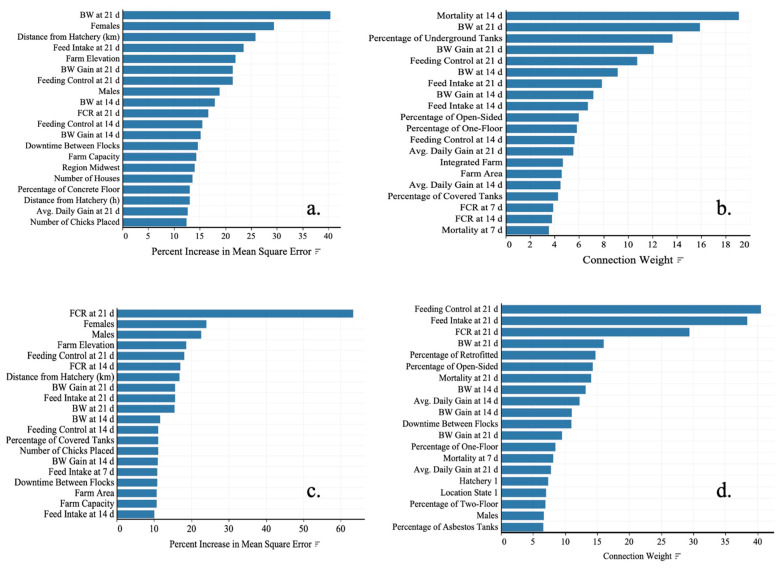
Variable importance of (**a**) BW with RF, (**b**) BW with ANN, (**c**) FCR with RF, and (**d**) FCR with ANN at 35 d on Ross 308 AP chickens raised under commercial tropical conditions.

**Figure 4 animals-13-03312-f004:**
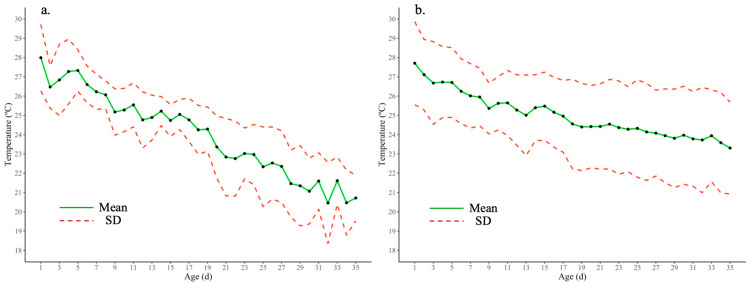
Mean ± SD of house temperature from flocks that presented (**a**) more than 450 h (53.6% of the time) within the temperatures recommended by Aviagen [38] for chickens between 1 and 35 d and (**b**) flocks affected by cumulative exposure to temperatures 2.5 °C lower than recommended. These graphics demonstrate the higher variability or wider confidence interval observed in houses (**b**) with flocks with more cumulative hours below target.

**Table 1 animals-13-03312-t001:** Descriptive statistics for farm-associated categorical variables.

Variable	Category	Northwest		Midwest		East	
		*n*	%	*n*	%	*n*	%
		--------------------------------------- (farms ^1^) ---------------------------------------					
Farms		42	48.84	17	19.77	27	31.40
Technification level ^2^	High	7	8.14	9	10.47	10	11.63
	Low	35	40.70	8	9.30	17	19.77
Chick source	Hatchery 1	42	48.84	0	0.00	0	0.00
	Hatchery 2	0	0.00	17	19.77	27	31.40
Litter type	Rice hulls	0	0.00	1	1.16	27	31.40
	Wood shavings	42	48.84	16	18.60	0	0.00
		--------------------------------------- (houses ^3^) ---------------------------------------					
Houses		367	49.80	153	20.76	217	29.44
Type of house	Open-sided	358	48.31	118	15.92	194	26.18
	Retrofitted	15	2.02	21	2.83	23	3.10
	Controlled	0	0.00	12	1.62	0	0.00
House stories ^4^	1	262	35.74	143	19.51	217	29.60
	2	101	13.78	10	1.36	0	0.00
House floor type	Soil	273	37.24	61	8.32	96	13.10
	Concrete	90	12.28	92	12.55	121	16.51
Water storage system	Covered	202	27.41	118	16.01	148	20.08
	Uncovered	157	21.30	41	5.56	69	9.36
	Underground	2	0.27	0	0.00	0	0.00

^1^ Variable percentage based on the number of farms by factor level over the total number of farms (*n* = 86); ^2^ technification levels. High level = farms with greater capacity, more houses, and more retrofitted houses; low = farms with less capacity and few houses with open-sided structures; ^3^ variable percentage based on the number of houses by factor level over the total number of houses (*n* = 737). Number of houses in each farm vary between 1 and 22. ^4^ Number of houses with one or two stories or levels enabled to place chicks.

**Table 2 animals-13-03312-t002:** Descriptive statistics for farm-associated continuous variables.

Variable	Northwest				Midwest				East			
	Mean	SD	Min	Max	Mean	SD	Min	Max	Mean	SD	Min	Max
Altitude (m.a.s.l.)	1588	398	1050	2353	1184	300	915	1955	1184	466	99	2253
Litter reuse cycles (# times)	1.9	1.85	0	12	5.58	4.15	0	12	0.44	0.57	0	2
Distance from the hatchery (km)	39.1	18.6	5	80	188.8	29.4	132	245	444.6	40	358	531
Distance from the hatchery (h)	1.5	0.6	0.1	3	5.4	0.7	4	6.5	8.3	1.1	6	11
Downtime between flocks (d)	14.1	0.4	14	16	16.3	1.4	14	21	13.0	1.6	11	20
Farm area (m^2^)	6435	4607	1620	24,992	12,655	8761	1440	31,680	7795	9238	1240	48,956

SD = standard deviation; Min = minimum value; and Max = maximum value. # = average number of times that litter was used.

**Table 3 animals-13-03312-t003:** Correlation coefficients (*r*) and regression estimates of live performance parameters of Ross 308 AP broilers from 0 to 28 d and hours of exposure to a relative humidity below and above the recommended values by Aviagen [38].

Age	Live Performance	Relative Humidity (%)	Age of Exposure (d)	*r*	*p*-Value Correlation	*n*	Intercept	Estimate	R^2^	R^2^ Adj	RMSE	*p*-Value LinearRegression
7	FCR	<50	0 to 7	−0.51	<0.001	46	0.936	−0.002	0.26	0.24	0.06	<0.001
21	Week mortality	<50	14 to 21	−0.46	0.015	28	0.155	−0.004	0.21	0.18	0.07	0.015
21	Cumulative mortality	<50	14 to 21	−0.45	0.015	29	0.209	−0.003	0.20	0.17	0.06	0.015
21	FCR	<50 or >75	14 to 21	0.41	<0.001	69	1.277	0.001	0.17	0.16	0.06	<0.001
28	FCR	<50	21 to 28	0.41	0.035	27	1.360	0.003	0.17	0.13	0.06	0.035
28	FCR	>75	0 to 28	0.41	0.001	68	1.384	0.0002	0.17	0.16	0.06	0.001
28	FCR	<50 or >75	0 to 28	0.41	<0.001	70	1.372	0.0002	0.17	0.16	0.06	<0.001

FCR = feed conversion ratio at each age. *n* = 80.

**Table 4 animals-13-03312-t004:** Effect of farm-associated factors on BW, FCR, and cumulative mortality at 35 d of male and female Ross 308 AP broilers raised under commercial tropical conditions.

Item	Category/Cluster Mean	SD	Males				Females			
			*n*	BW	FCR	Mortality	*n*	BW	FCR	Mortality
			Flocks	--(g)--	-(g/g)-	-- (%) --	flocks	-- (g) --	-(g/g)-	--(%)--
Region	Northwest		381	1836 ^c^	1.461 ^c^	2.98 ^a^	385	1729 ^b^	1.514 ^c^	2.68 ^ab^
	Midwest		148	1947 ^a^	1.493 ^b^	2.52 ^b^	154	1751 ^a^	1.533 ^b^	2.27 ^b^
	East		669	1882 ^b^	1.515 ^a^	2.85 ^a^	670	1673 ^c^	1.557 ^a^	2.71 ^a^
	SEM ±			4	0.003	0.10		4	0.003	0.09
	CV %			4.1	3.3	26.6		4.3	2.9	27.3
Technification level	High		497	1889 ^a^	1.507 ^a^	2.94	502	1690 ^b^	1.550 ^a^	2.72 ^a^
	Low		687	1866 ^b^	1.487 ^b^	2.79	694	1708 ^a^	1.533 ^b^	2.54 ^b^
	SEM ±			3	0.002	0.07		3	0.002	0.07
	CV %			4.4	3.6	26.8		4.7	3.1	28.2
Altitude (m.a.s.l.)	1041	349	376	1873 ^b^	1.522 ^b^	2.69 ^b^	359	1730 ^b^	1.519 ^b^	1.96 ^c^
	1446	498	172	1792 ^c^	1.577 ^a^	3.82 ^a^	645	1660 ^c^	1.573 ^a^	3.06 ^a^
	1670	434	650	1898 ^a^	1.457 ^c^	2.71 ^b^	205	1774 ^a^	1.475 ^c^	2.43 ^b^
	SEM ±			4	0.003	0.09		4	0.002	0.09
	CV %			4.0	2.2	26.1		3.9	2.0	27.4
* **Source of variation** *			**-------------------------------------- *p*-value --------------------------------------**							
Region				<0.001	<0.001	0.019		<0.001	<0.001	0.003
Type of administration				<0.001	<0.001	0.087		<0.001	<0.001	0.008
Altitude				<0.001	<0.001	<0.001		<0.001	<0.001	<0.001
Litter type	Rice hulls		672	1883 ^a^	1.514 ^a^	2.85	676	1674 ^b^	1.556 ^a^	2.69 ^a^
	Wood shaving		526	1867 ^b^	1.470 ^b^	2.87	533	1733 ^a^	1.519 ^b^	2.54 ^b^
	SEM ±			3	0.002	0.07		3	0.002	0.07
	CV %			4.4	3.4	26.8		4.3	2.9	28.9
Litter reuse cycles number	0.8	0.7	348	1819 ^b^	1.556 ^a^	3.47 ^a^	797	1671 ^c^	1.561 ^a^	2.89 ^a^
	1.3	1.2	767	1898 ^a^	1.465 ^c^	2.59 ^c^	297	1769 ^a^	1.481 ^c^	2.10 ^b^
	5.8	2.2	32	1898 ^a^	1.564 ^a^	3.41 ^ab^	69	1722 ^b^	1.562 ^a^	2.10 ^b^
	12.0	0.0	51	1898 ^a^	1.479 ^b^	2.53 ^bc^	46	1718 ^b^	1.522 ^b^	2.24 ^b^
	SEM ±			8	0.004	0.17		6	0.003	0.15
	CV %			4.0	2.4	26.0		4.0	2.3	28.1
Downtime between flocks (d)	12.7	1.0	314	1822 ^c^	1.562 ^a^	3.44 ^a^	678	1662 ^c^	1.569 ^a^	2.94 ^a^
	13.5	1.2	662	1906 ^a^	1.457 ^c^	2.51 ^b^	150	1778 ^a^	1.466 ^c^	2.05 ^b^
	15.0	1.6	222	1859 ^b^	1.510 ^b^	3.11 ^a^	381	1735 ^b^	1.517 ^b^	2.31 ^b^
	SEM ±			4	0.002	0.09		4	0.002	0.09
	CV %			4.0	2.2	25.9		3.9	2.1	28.1
* **Source of variation** *			**---------------------------------------- *p*-value -------------------------------------**							
Litter type				0.001	<0.001	0.657		<0.001	<0.001	0.011
Litter reuse cycle number				<0.001	<0.001	<0.001		<0.001	<0.001	<0.001
Downtime between flocks				<0.001	<0.001	<0.001		<0.001	<0.001	<0.001

^a–c^ Means in columns followed by different superscript letters are statistically different with Tukey’s or student’s *t*-test (*p* < 0.05). BW = body weight; FCR = feed conversion ratio. m.a.s.l. = meters above sea level.

**Table 5 animals-13-03312-t005:** Effect of chick source and hatchery distance on BW, FCR, and cumulative mortality at 35 d of male and female Ross 308 AP broilers raised under commercial tropical conditions.

Item	Category/Cluster Mean	SD	Males				Females			
			*n*	BW	FCR	Mortality	*n*	BW	FCR	Mortality
			Flocks	--(g)--	-(g/g)-	-- (%) --	Flocks	-- (g) --	-(g/g)-	--(%)--
Chick source	Hatchery 1		381	1836 ^b^	1.461 ^b^	3.00	385	1729 ^a^	1.514 ^b^	2.65
	Hatchery 2		817	1894 ^a^	1.511 ^a^	2.79	824	1687 ^b^	1.552 ^a^	2.62
	SEM ±			3	0.003	0.07		3	0.002	0.07
	CV %			4.2	3.4	26.8		4.5	3.0	0.1
Distance from hatchery (km)	73	71	461	1875 ^b^	1.455 ^c^	2.71 ^b^	322	1758 ^a^	1.491 ^c^	2.18 ^b^
	356	169	257	1812 ^c^	1.572 ^a^	3.54 ^a^	722	1665 ^c^	1.570 ^a^	2.92 ^a^
	454	64	480	1909 ^a^	1.492 ^b^	2.65 ^b^	165	1740 ^b^	1.503 ^b^	2.21 ^b^
	SEM ±			4	0.002	0.09		4	0.002	0.09
	CV %			4.0	2.3	26.2		3.9	2.0	28.1
Distance from hatchery (h)	2.1	1.6	429	1873 ^a^	1.449 ^c^	2.73 ^b^	258	1755 ^a^	1.486 ^c^	2.21 ^b^
	7.0	2.9	247	1808 ^b^	1.573 ^a^	3.58 ^a^	714	1663 ^b^	1.571 ^a^	2.93 ^a^
	8.2	1.8	522	1909 ^a^	1.496 ^b^	2.63 ^b^	237	1750 ^a^	1.507 ^b^	2.16 ^b^
	SEM ±			4	0.002	3.85		4	0.002	3.85
	CV %			4.0	2.2	26.2		3.9	2.0	28.1
** *Source of variation* **			**---------------------------------------- *p*-value ----------------------------------------**							
Chick source				<0.001	<0.001	0.146		<0.001	<0.001	0.349
Distance from the hatchery (km)				<0.001	<0.001	<0.001		<0.001	<0.001	<0.001
Distance from the hatchery (h)				<0.001	<0.001	<0.001		<0.001	<0.001	<0.001

^a–c^ Means in columns followed by different superscript letters are statistically different with Tukey’s or student’s *t*-test (*p* < 0.05).

**Table 6 animals-13-03312-t006:** Variable contribution of MLR fit with growth and farm-associated conditions on BW and FCR at 35 d of Ross 308 AP chickens reared under commercial tropical conditions.

Variable	Sum Sq	Contribution (%) ^1^	*F*-Value	*p*-Value
**BW**				
Females	13,108,737.0	89.27	2325.7	<0.001
Feed Intake 14 d	646,918.0	4.41	114.8	<0.001
BW Gain at 7 d	267,502.8	1.82	47.5	<0.001
FCR at 21 d	209,950.5	1.43	37.2	<0.001
Percentage of Retrofitted	200,095.7	1.36	35.5	<0.001
Percentage of Open-Sided	119,749.4	0.82	21.2	<0.001
Region	79,587.1	0.54	14.1	<0.001
Percentage of Underground Tanks	51,800.2	0.35	9.2	0.002
**FCR**				
FCR at 21 d	0.41	42.41	162.21	<0.001
Downtime Between Flocks	0.18	18.67	71.41	<0.001
Feed Intake 14 d	0.17	17.63	67.43	<0.001
Percentage of Retrofitted	0.07	7.34	28.09	<0.001
Percentage of Underground Tanks	0.04	3.94	15.06	<0.001
Percentage of Open-Sided	0.03	2.69	10.29	0.001
Percentage of Concrete Floor	0.02	2.56	9.80	0.002
Feed Intake 21 d	0.02	2.22	8.51	0.004
Month at Placement	0.01	1.47	5.62	0.018
Mortality at 21 d	0.01	1.06	4.04	0.045

^1^ Variable contribution of MLR models expressed as the percentage of the variable sum of squares over the sum of squares of the model. Sum sq = Sum of squares; BW = body weight; FCR = feed conversion ratio.

## Data Availability

The data used in this study are the property of Grupo BIOS Inc., Colombia, and are, therefore, not publicly available.

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
