# Peer review of "Effect of Environmental and Farm-Associated Factors on Live Performance Parameters of Broilers Raised under Commercial Tropical Conditions"

_animals, 2023, doi:10.3390/ani13213312_

Round 1
Reviewer 1 Report
It is indeed a significant study to elucidate the impact of temperature, relative humidity, and farm-specific factors on broiler live performance, particularly when correlating broiler age with environmental conditions and determining which farm factors contribute most to broiler performance. To enhance the entire paper, there are several areas that require further development:
1. Introduction (lines 96-103): The challenges of utilizing commercial data need to be explored more thoroughly. Rather than merely stating it as a challenge, provide more substance and references to explain why this is the case. Similarly, the reasoning behind why machine learning (ML) could be a potential solution for analyzing environmental data needs further explanation.
2. Data Analysis (lines 172-178): The introduction to k-means clustering should be linked with the farm management factor for better coherence. In line 223, the methodology to calculate R2 and RMSE should be explained, including their respective equations. Given that it's their first appearance in the paper, their full names should be mentioned.
3. Discussion (lines 523-526): The paper refers to "variable importance analyses from the supervised ML models". It would be beneficial to provide more detailed information on what these analyses entail.
4. Section 4.3 and Data Analysis 3 (line 207): The three machine learning methods being utilized should be clearly specified at the outset. The citation of a broiler management handbook (reference 37) seems out of place and needs justification. Moreover, in section 4.3, you've discussed Multiple Linear Regression (MLR) and Random Forest (RF) models. However, the comparison of the Artificial Neural Network (ANN) model with these two is not adequately covered and should be included for a comprehensive evaluation.
Addressing these areas will significantly enhance the clarity and depth of your work.
Author Response
Introduction (lines 96-103): The challenges of utilizing commercial data need to be explored more thoroughly. Rather than merely stating it as a challenge, provide more substance and references to explain why this is the case. Similarly, the reasoning behind why machine learning (ML) could be a potential solution for analyzing environmental data needs further explanation.
Answer. Thanks for your comment. Additional details about the challenges of using commercial data were added to lines 96-103, and 105 to 117.
Data Analysis (lines 172-178): The introduction to k-means clustering should be linked with the farm management factor for better coherence.
Answer. Thanks for your comment. A link of the variables clustered were used as a link with the management conditions. Lines 205-212.
In line 223, the methodology to calculate R2 and RMSE should be explained, including their respective equations. Given that it's their first appearance in the paper, their full names should be mentioned.
Answer. Thanks for your comments. R2 and RMSE are basic statistic concepts and widely employed goodness-of-fit metrics in statistics. Then, the authors consider it is not necessary to be further explained. A reference was added to the line 264.
3. Discussion (lines 523-526): The paper refers to "variable importance analyses from the supervised ML models". It would be beneficial to provide more detailed information on what these analyses entail.
Answer. Thanks for your comment. A description of variable importance analysis was added to lines 269-278.
4. Section 4.3 and Data Analysis 3 (line 207): The three machine learning methods being utilized should be clearly specified at the outset.
Answer. Thanks for your comment. A clear statement of the methods used in this study was added to line 246-250.
The citation of a broiler management handbook (reference 37) seems out of place and needs justification.
Answer. Thanks for your comment. Reference justified.
Moreover, in section 4.3, you've discussed Multiple Linear Regression (MLR) and Random Forest (RF) models. However, the comparison of the Artificial Neural Network (ANN) model with these two is not adequately covered and should be included for a comprehensive evaluation.
Answer. Thanks for your comment. ANN was included in the discussion, lines 606-618. References and potential benefits of ANN as predictive tool were added.
Reviewer 2 Report
Overall, this study underscores the importance of understanding and managing environmental, management, and housing factors in commercial poultry operations to optimize broiler chicken performance. Machine learning techniques, such as random forest and artificial neural networks, are employed to reveal the complex relationships between various factors and their impact on broiler growth and performance.
There could be many confounding factors affecting the results of this study. How you treated those factors? The abbreviations must be defined below the tables. Conclusion must be provided in the abstract
Author Response
There could be many confounding factors affecting the results of this study. How you treated those factors?
Answer. Thanks for your comment. The importance of confounding factors and the benefits of ML to deal with these confounding variables was discussed in lines 101 to 117. The RF and ANN are less affected by confounding variables since they iteratively run multiple analysis with different variables so the confounding variables are intrinsically removed from the modeling process.
The abbreviations must be defined below the tables.
Answer. Thanks for your comment. Abbreviations were added to tables.
Conclusion must be provided in the abstract
Answer. Thanks for your comment. The main conclusions were added to the abstract.
Reviewer 3 Report
Overall, this is an interesting paper and well written. It is a metanalysis of a large dataset obtained from a single broiler integrator. I found the Data Analysis 2 and 3 portions of the paper to be interesting. However, I feel the performance and environment portion of the paper needs to be bolstered some to make the data more useful and relevant. If it can't be bolstered, I would consider removing it from the paper.
My primary concern mostly relates to temperature portion of the "performance and environment" part of the paper. Line 25 states that exposure time to lower or greater than 1.5, 2.5, 5.0, and 7 C from the recommendations were calculated. However, Figures 1 and 2 only show data for the number of hours that the temperature is either 1.5, 2.5, or 5 C below the recommended target. Nothing is really mentioned about the influence of temperatures above target, which lead to heat stress and performance declines.
Is Colombia a heating dominated climate or a cooling dominated climate? I would think cooling dominated, which makes me wonder why the focus is on temperatures below the recommended target.
Also, to put the temperature and performance data into context, a histogram of the number of cumulative hours that the flocks are above and below 1.5, 2.5, 5.0, and 7.0 would be very useful. A table of the number of flocks that experienced these conditions would also help. Figures 1 and 2 only have a about 17 data points in all the graphs. Of the roughly 2,600 flocks of data, are there only 15-20 that fit the criteria presented in the figures? It's evident that cold temperatures affect production, but are these conditions rare enough that they may not make that big of deal with regards to production. All this to say, wouldn't Colombia experience many more hours above the target that are going to have a much more drastic influence on production? I really think this needs to be addressed in a meaningful way or else I'm not sure this section provides much significance to the paper.
I would recommend changing the x-axis titles in Figure 1 to something more descriptive like "number of hours temperature is 5C below target from...".
Please clarify lines 112-114. Did you monitor data on these farm from January to July 2020. This is a little confusing because the dataset you use is from 2018-2020.
Of, the 1,347 male and 1,353 flocks for which you got data, how many were included in you analysis?
Line 125: Was this data taken directly from a controller or were external sensors and logger installed on the farms? I'm assuming data is from controllers, which is not made clear.
Houses usually have more than one sensor. Did you include data from all the sensors in your analysis.
Can you address the accuracy of the sensors and how this may influence your results? For example, sensors and controllers are not always calibrated and can read temperatures that are not accurate.
What is the range of house sizes?
I found figure 4 confusing. Figure 4b is showing cumulative exposure to temps lower than 2.5C than recommended but the temperature are higher than those presented in Figure 4a. Also, I would consider rephrasing to "cumulative exposure to to temps greater than 2.5C below the target."
Line 541 states "temperature lower than 2.5C added up to the variable..."
Author Response
My primary concern mostly relates to temperature portion of the "performance and environment" part of the paper. Line 25 states that exposure time to lower or greater than 1.5, 2.5, 5.0, and 7 C from the recommendations were calculated. However, Figures 1 and 2 only show data for the number of hours that the temperature is either 1.5, 2.5, or 5 C below the recommended target. Nothing is really mentioned about the influence of temperatures above target, which lead to heat stress and performance declines.
Answer. Thanks for your comment. In fact, the origin of the datasets and size of each dataset was not clearly defined. We added details in materials and methods, results, and discussion. Data from environmental monitoring only came from 29 of the 86 farms and only 80 from 2,700 flocks. In the conclusion an statement indicates “Not enough flocks were exposed to temperatures above 1.5 oC in these datasets to detect effect of heat stress early in life.” was added in the conclusion. Additionally, a result related to temperatures above the target was added in the results lines 309-316.
Is Colombia a heating dominated climate or a cooling dominated climate? I would think cooling dominated, which makes me wonder why the focus is on temperatures below the recommended target.
Answer. Thanks for your comment. Details of the origin of the environmental data were added in lines 31-34, 139-144 , and 288-289. Colombia is located in an Andean tropical region and the weather depends on the altitude. The temperature decreases as altitude increases. A statement about weather conditions was added to lines 447-452.
Also, to put the temperature and performance data into context, a histogram of the number of cumulative hours that the flocks are above and below 1.5, 2.5, 5.0, and 7.0 would be very useful. A table of the number of flocks that experienced these conditions would also help. Figures 1 and 2 only have a about 17 data points in all the graphs. Of the roughly 2,600 flocks of data, are there only 15-20 that fit the criteria presented in the figures? It's evident that cold temperatures affect production, but are these conditions rare enough that they may not make that big of deal with regards to production. All this to say, wouldn't Colombia experience many more hours above the target that are going to have a much more drastic influence on production? I really think this needs to be addressed in a meaningful way or else I'm not sure this section provides much significance to the paper.
Answer. Thanks for your comment. A statement about the flocks and farms subject to environmental monitoring was added to lines 140-141 and emphasized in lines 158-159 and 282-283.
I would recommend changing the x-axis titles in Figure 1 to something more descriptive like "number of hours temperature is 5C below target from...".
Answer. Thanks for your comment. x-axis corrected in all figures as advised.
Please clarify lines 112-114. Did you monitor data on these farm from January to July 2020. This is a little confusing because the dataset you use is from 2018-2020.
Answer. Thanks for your comment. Time frame for environmental monitoring detailed in lines 140-141
Of, the 1,347 male and 1,353 flocks for which you got data, how many were included in you analysis?
Answer. Thanks for your comment. For environmental analysis 80 out of 2,700 that were subject for environmental monitoring. For management conditions and machine learning models all 2,700 flocks. Number of flocks for each analysis was detailed in materials and methods and explained in results and discussion.
Line 125: Was this data taken directly from a controller or were external sensors and logger installed on the farms? I'm assuming data is from controllers, which is not made clear.
Answer. Thanks for your comment. Data was taken from external sensors installed within broiler houses. A detail about this procedure was added to lines 147-152.
Houses usually have more than one sensor. Did you include data from all the sensors in your analysis.
Thanks for your comment. A detail about this procedure was added to lines 146-153.
Can you address the accuracy of the sensors and how this may influence your results? For example, sensors and controllers are not always calibrated and can read temperatures that are not accurate.
Answer. Thanks for your comment. Sensors were provided by a company dedicated to analytics in animal production. A detail about biased data was added in lines 140-144
What is the range of house sizes?
Answer. Thanks for your question. The range was added to lines 128 to 129.
I found figure 4 confusing. Figure 4b is showing cumulative exposure to temps lower than 2.5C than recommended but the temperature are higher than those presented in Figure 4a. Also, I would consider rephrasing to "cumulative exposure to temps greater than 2.5C below the target."
Answer. Thanks for your comment. x-axis corrected in all figures as advised.
Line 541 states "temperature lower than 2.5C added up to the variable..." Answer.Detail corrected.
Round 2
Reviewer 1 Report
After carefully reviewing all the contents, all my previous comments have been resolved. Therefore, I don't have any further comments now; the paper is ready to be published.